# Circulation of hepatitis delta virus and occult hepatitis B virus infection amongst HIV/HBV co-infected patients in Korle-Bu, Ghana

**Keren Attiku**[1,2], **Joseph Bonney**[1,2]*, **Esinam Agbosu**[1,2], **Evelyn Bonney**[1,2], **Peter Puplampu**[3], **Vincent Ganu**[3], **John Odoom**[1,2], **James Aboagye**[2], **John Mensah**[3], **Seth Agyemang**[1,3], **Yaw Awuku-Larbi**[4], **Augustina Arjarquah**[1,2], **Gifty Mawuli**[2], **Osbourne Quaye**[1]*

**1** West African Centre for Cell Biology of Infectious Pathogens, Department of Biochemistry, Cell and Molecular Biology, University of Ghana, Accra, Ghana, **2** Department of Virology, Noguchi Memorial Institute for Medical Research, University of Ghana, Accra, Ghana, **3** Fevers Unit, Korle-Bu Teaching Hospital, Accra, Ghana, **4** School of Public Health, University of Witwatersrand, Johannesburg, South Africa

* kbonney@noguchi.ug.edu.gh (JB); oquaye@ug.edu.gh (OQ)

## Abstract

### Background

Within HIV/HBV infected patients, an increase in HDV infection has been observed; there is inadequate information on HDV prevalence as well as virologic profile in Ghana. This study sought to determine the presence of HDV in HIV/HBV co-infected patients in Ghana.

### Methods

This was a longitudinal purposive study which enrolled 113 HIV/HBV co-infected patients attending clinic at Korle-Bu Teaching Hospital (KBTH) in Accra, Ghana. After consenting, 5 mL whole blood was collected at two-time points (baseline and 4–6 months afterwards). The sera obtained were tested to confirm the presence of HIV, HBV antibodies and/or antigens, and HBV DNA. Antibodies and viral RNA were also determined for HDV. Amplified HBV DNA and HDV RNA were sequenced and phylogenetic analysis carried out with reference sequences from the GenBank to establish the genotypes.

### Results

Of the 113 samples tested 63 (55.7%) were females and 50 (44.25%) were males with a median age of 45 years. A total of 100 (88.5%) samples had detectable HBV surface antigen (HBsAg), and 32 out of the 113 had detectable HBV DNA. Nucleotide sequences were obtained for 15 and 2 samples of HBV and HDV, respectively. Phylogenetic analysis was predominantly genotype E for the HBVs and genotype 1 for the HDVs. Of the 13 samples that were HBsAg unreactive, 4 (30.8%) had detectable HBV DNA suggesting the incidence of occult HBV infections. The percentage occurrence of HDV in this study was observed to be 3.54.

**Data Availability Statement:** All relevant data are within the manuscript and its Supporting Information file.

**Funding:** KA received funding support through the West African Centre for Cell Biology of Infectious Pathogens (WACCBIP) under the World Bank's African Centres of Excellence (ACE) Project (https://www.waccbip.org) and the WHO supported National Poliomyelitis Reference Laboratory, Ghana. The funders had no role in the study design, data collection and analysis, or preparation of manuscript.

**Competing interests:** The authors have declared that no competing interests exist.

## Conclusion

Our data suggest the presence and circulation of HDV and incidence of occult HBV infection in HIV/HBV co-infected patients in Ghana. This informs health staff and makes it imperative to look out for the presence of HDV and occult HBV in HIV/HBV co-infected patients presenting with potential risk of liver cancers and HBV transmission through haemodialysis and blood transfusions.

## Introduction

Hepatitis delta virus (HDV) is a defective virus that infects only persons with Hepatitis B virus (HBV) infection [1]. The D virus infections are known to be acquired through parenteral and sexual routes as well as through transfusions and injections, which are also important routes for human immunodeficient virus (HIV) and HBV transmissions [2]. Co-infection of HDV and HBV has been shown to worsen the severity of acute hepatitis and hepatic decompensation, with an associated increase in liver related mortality [3], with HDV super-infections also adding considerably to the high burden of chronic liver diseases. A previously stable chronic carrier of HBV can rapidly progress to cirrhosis within two years after a super-infection with HDV [4]. Globally, it is estimated that about 20 million people infected with HBV are also infected with HDV [5], and 7.33% prevalence of HBV/HDV has been reported for West Africa [6]. Meanwhile, a study in Mongolia has reported that 60% of HBV infected patients also have HDV infection [7]. Thus, there is a big gap in the report of HBV/HDV prevalence in many countries, since HBV-infected patients are not normally tested for HDV [1].

Hepatitis delta virus in HIV/HBV co-infected patients leads to accelerated hepatic disease progression with higher rates of liver cirrhosis and liver related mortality as well as complications in treatment possibilities compared with HDV in HBV mono-infected patients [8]. In addition, current infection with HDV complicates viral treatment as regimen against HBV does not affect HDV [1]. Though prevention of HBV results in the prevention of HDV, treatment of HBV/HDV coinfected patients differs from that of mono HBV infection [1].

Hepatitis D virus infection has emerged as a significant public health concern throughout the world, with continuous increase in the rate of HDV infection in HIV/HBV co-infected patients [9]. With unknown information of deaths due to HBV with HDV as a cofactor [10], as well as limited information on HDV prevalence and virologic profile among HIV patients [11] both in Ghana and globally, it has become necessary to do further investigations into the prevalence, biochemical indices and disease severity of HBV/HDV co-infection in HIV patients.

## Methods

### Patients

This was a longitudinal purposive study which enrolled HIV/HBV co-infected patients who attended clinic at the Fever's Unit of the Korle-Bu Teaching Hospital (KBTH) in Accra, Ghana from September 2018 to May 2019. The study was approved by the institutional review boards of Noguchi Memorial Institute for Medical Research (NMIMR-IRB CPN 090/17-18) and KBTH (KBTH/STC/IRB/000113/2018). Patients were identified and selected by their HIV and HBV status from their hospital folders. The HIV patients whose HBV status were negative or had previously received HBV vaccine were not recruited for this study. Written informed

consent was obtained from eligible patients before whole blood was taken and separated into serum. The whole blood was taken from one hundred and thirteen (113) patients at the beginning of the study, and a followed-up sample was taken after 4 to 6 months after the initial enrolment. Demographic and clinical details of the patients were obtained with the use of a structured questionnaire.

## Laboratory analysis

The HIV status of all enrolled patients were confirmed using OraQuick HIV 1/2 Rapid Antibody Test (OraSure Technologies, PA, USA), and HIV RNA load was determined using the Roche Cobas Ampliprep/Taqman qRT-PCR assay, v.2.0 Roche Diagnostics, Indianapolis, IN, USA) following manufacturer's protocol and interpretation. The HBV status of all the 113 patients was confirmed first with the Alere Determine (Alere Medical Co. Ltd., Tokyo, Japan) rapid test kits, and for the samples with enough serum, a second confirmatory test was done using an electrochemiluminescence assay (Elecsys HBsAg II assay from Roche Diagnostics, Hamburg, Germany) following manufacturer's protocol. The biochemistry analysis for Alanine aminotransferase (ALT) and Aspartate aminotransferase (AST) was quantitated using the VITROS ALT slide method on the VITROS 5600 integrated system according to manufacturer's instructions (Ortho Clinical Diagnostics, Rochester, NY, USA).

## HDV screening

All 113 samples were screened for anti-HDV IgG antibodies using hepatitis D virus antigen antibody (IgG) ELISA Kit (Colorimetric) from Bio-TechneTechne, Abingdon, UK, by following the manufacturer's protocol and interpretation of results strictly. The precision in terms of coefficient of variation for this kit was stated to be <15%.

## Detection and quantification of HBV DNA and HDV RNA

Detection and quantification of HBV DNA and HDV RNA targeting the core regions of each virus were done using a commercial kit (QPCR Kit, DNA, Hepatitis B virus and QPCR Kit, RNA, Hepatitis Delta virus, Bibby Scientific, Cyprus) with slight modifications to the manufacturer's pipetting protocol. Reaction mixtures for detection of HDV and HBV contained 5.0 μL 2 x qPCR master mix (Quanta Tough mix), 2.0 μL of nuclease free water, 0.5 μL of HDV primer/probe mix and 5 μL of template DNA in a total volume of 12.5 μL per sample. The amplification conditions for HDV detection were 10 minutes at 55˚C reverse transcription stage, 2 minutes at 95˚C to stimulate the Taq polymerase enzyme, and 50 cycles of 10 seconds at 95˚C and 1 minute at 60˚C, whilst the amplification conditions of HBV were 95˚C for 2 minutes and 50 cycles of 95˚C for 10 seconds and 1 minute at 60˚C. To validate the results, a separate reaction mix was prepared as a control using an endogenous primer/probe mix to detect the presence of nucleic acid material in the extracted sample. The ABI 7500 PCR system, (Life technologies, Malaysia) was used in the detection of the HBV DNA / HDV RNA. The limit of detection for HBV DNA and HDV RNA according to manufacturer's protocol was 2 copies per microliter, whilst the specificity and sensitivity for HBV DNA and HDV RNA was stated to be 100% and 95% as well as 95% and >95% respectively.

## Sanger sequencing and phylogenetic analysis of HBV S-gene

All real-time PCR positives for HBV were amplified using a nested PCR protocol targeting the S-gene as described previously [12]. The primer sequences as well as the range they target have been listed in Table 1. The amplicon size obtained after the nested PCR is ~400 base pairs (bp).

**Table 1. Primer sequences used in amplifying and sequencing the S-gene of HBV.**

| Region | Primer name | Primer position on the S gene | Nucleotide sequence of the primers (5´–3´) |
|---|---|---|---|
| SI | SIF | 179s | CTAGGACCCCTGCTGGTGTT |
| | SIR | 682as | TCGAACCACTGAACAAATGGCACT |
| SN | SNF | 216s | GTTGACAAGAATCCTCACAATACC |
| | SNR | 639as | GGCTGAGGCCCACTCCCATA |

Sanger sequencing of the amplicons obtained from the nested PCR was done by adapting the pipetting protocol of the Ghana National Polio Reference Laboratory. Briefly, a mixture of 2 μL of Big Dye v3.1, 1 μL of 5 X sequencing buffer, 1 μL primer, 5 μL water and 1 μL 50 ng template DNA was prepared and sequenced with the following thermal conditions 96˚C for 3 minutes, 96˚C, 30 seconds; 55˚C, 30 seconds; 60˚C, 1 minute and a final extension of 60˚C, 10 minutes.

Phylogenetic analysis of the HBV sequences was completed with Bioedit software version 7.2.5 [13] and Molecular Evolutionary Genetics Analysis software version X (MEGA X) [14].

### Sanger sequencing and phylogenetic analysis of HDV R0-region

All real-time PCR and anti-HDV IgG antibody positives for HDV were amplified using a nested PCR protocol targeting the 3' R0 region as described previously [15]. The primer sequences as well as the range they target have been listed in Table 2. The amplicon size obtained after the nested PCR is 400 bp.

Sanger sequencing of the amplicons obtained from the nested PCR was done by adapting the pipetting protocol of the Ghana National Polio Reference Laboratory as described above.

Phylogenetic analysis of the sequence reads was completed using the Bioedit software version 7.2.5 [13] and Molecular Evolutionary Genetics Analysis software version X (MEGA X) [14].

### Statistical analysis

Demographic data were expressed in terms of means, medians and standard deviations. The students' T-test was used in testing for the association between liver enzyme levels and viral infection type. A univariate logistic regression analysis was used to test for association between liver enzyme activity, an indicator for liver damage and HBV viral load suppression, as well as other risk factors using STATA 15 software.

## Results

### Study participants

A total of 113 HIV/HBV patients were recruited from the Fever's Unit of the KBTH based on their hospital records and they were screened for HDV infection. All the patients recruited were retested by serological analyses for their coinfection status. Majority of the participants were females (56%). The median age of the patients was 45 years, and the age ranged from 24 to 73 years. A higher proportion of the patients (66%) were within the age range of 31–50 years

**Table 2. Primer sequences used in amplifying and sequencing the 3' R0 region of HDV RNA.**

| Region | Primer name | Nucleotide sequence of the primers (5´–3´) |
|---|---|---|
| R0 | 889s (R0F) | CATGCCGACCCGAAGAGGAAAG |
| (889–1289) | 1289as (R0R) | GAAGGAAAGGCCCTCGAGAACAAGA |

(Table 3). Almost all the patients (~98%) were on antiretroviral therapy (ART) except 2 (~2%) who were naïve. Thirty-seven of the patients have been on ART for 2 years or less, and ~92% of those on ART were on first line class of drugs which includes a combination of 2 nucleo(t) side reverse transcriptase inhibitor (NRTI) and 1 non-nucleo(t)side reverse transcriptase inhibitor (NNRTI) (Fig 1). All the patients receiving ART except for the 2 ART naïve patients are on lamivudine-based therapy with 30 of these also receiving tenofovir as part of their ART regimen (Table 3).

Few of the patients (~8%) were on second line class of drugs which includes a combination of 2 nucleo(t)side reverse transcriptase inhibitor (NRTI) and a Protease inhibitor. All study patients had not received hepatitis B vaccination (Table 1).

## HIV status

The HIV status of the study participants were re-tested using rapid diagnostic test kits (OraQuick HIV 1/2 Rapid Antibody Test, OraSure Technologies, Inc, Thailand) on baseline samples. All the 113 patients (100%) were reactive to the HIV rapid diagnostic test (Table 4). The viral load for HIV was also measured for all baseline samples and for patients who reported for follow ups. At baseline, 47 (~42%) of the patients had undetectable viral RNA copies, 28 (~25%) patients had RNA copies less than 20 cp/mL, 36 (~32%) patients had viral RNA copies greater than 20 cp/mL with a median of 7941.5 (21–1265314) cp/mL and 2 (~2) patients viral load could not be determined due to failed tests. At follow up, 48 out of the 113 patients enrolled at baseline were lost to follow up. With the 65 patients who had follow up samples taken, 26 had undetectable viral load, 14 had HIV load less than 20 cp/mL, 22 had viral copies greater than 20 cp/mL with a median age of 111 (22–493032) cp/mL out of which 15 had increased viral loads and 7 had reduced viral loads from baseline to follow up. Three patient samples failed the assay procedure (Table 4).

## HBV status

As a confirmation, a total of 100 (88.5%) out of the 113 study participants were reactive to hepatitis B surface antigen (HBsAg) on the Alere Determine rapid diagnostic test kit. HBsAg was

**Table 3. Demographic characteristics of study participants.**

| Gender | N = 113 |
|---|---|
| *Female* | 63 (55.8) |
| *Male* | 50 (44.2) |
| **Age (years)** | |
| *21–30* | 8 (7.1) |
| *31–50* | 75 (66.4) |
| *≥51* | 30 (26.6) |
| *Median (IQR)* | 45 (24–73) |
| **Duration of ART (months)** | |
| *0–24* | 38 (33.6) |
| *≥25* | 75 (66.4) |
| **Vaccination** | |
| *No* | 113 (100) |
| *Yes* | 0 (0) |
| **Class of ART** | |
| *First Line (2NRTI+1NNRTI)* | 102 (92.0) |
| *Second Line (2NRTI + 1 PI)* | 9 (8.0) |

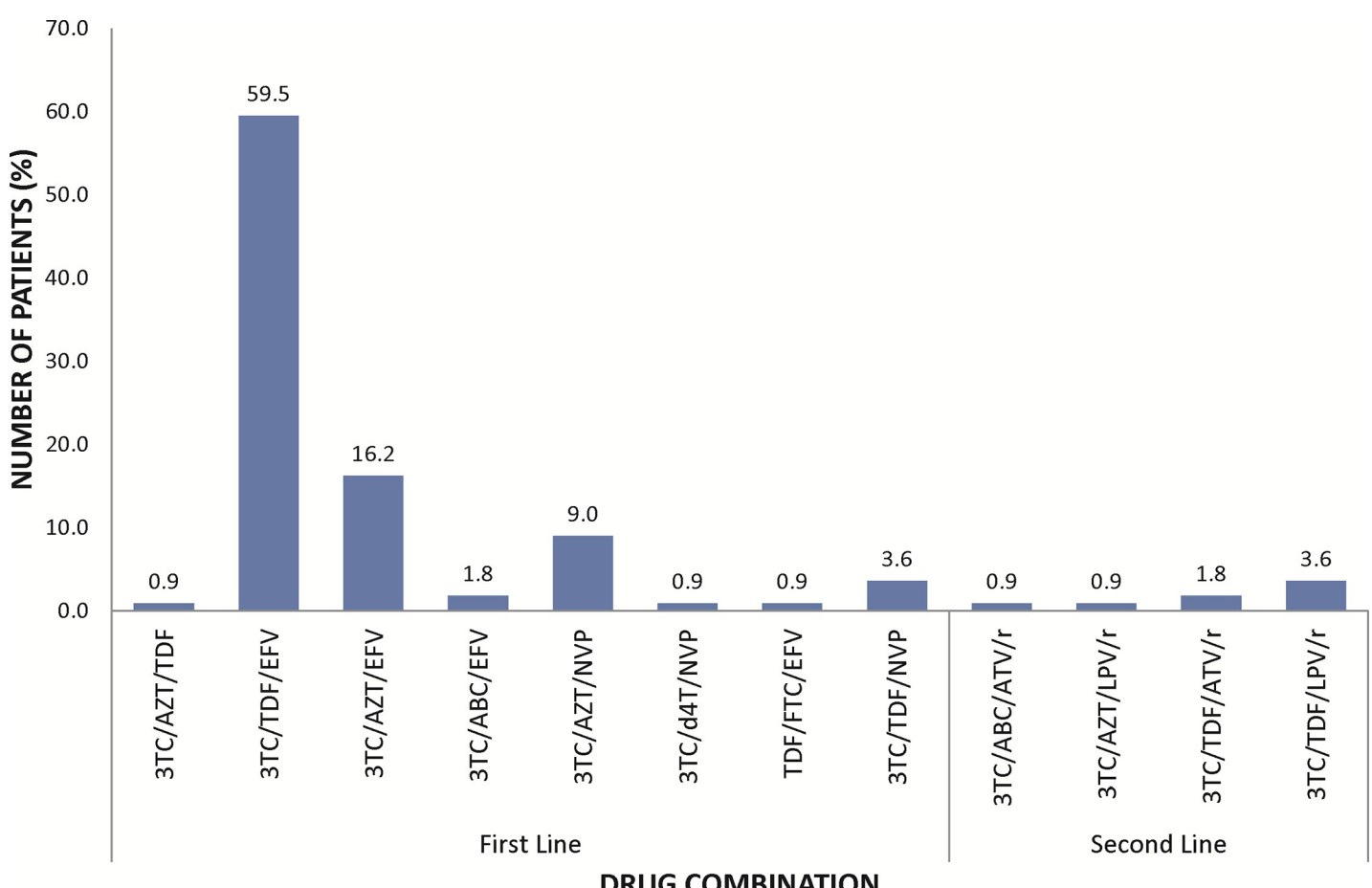

**Fig 1. Treatment options used in the management of HIV/HBV co-infected patients at the Fever's Unit of the Korle-Bu Teaching Hospital.** First line drugs (Lamivudine (3TC), abacavir (ABC), zidovudine (AZT), emtricitabine (FTC), stavudine (d4T) or tenofovir-DF (TDF), efavirenz (EFV) or nevirapine (NVP), second line class of drugs (atavanavir (ATV) or Lopinavir (LVP) boosted with ritonavir (/r).

quantified using the ECLIA kit with a calculated median of 1934 IU/mL, ranging from 48–9118 IU/mL. Using real time PCR assay to assess HBV load, 31 (27.4%) of the participants who were enrolled had detectable HBV DNA, with viral loads established for 13 (41.9%) out of the 31. The median HBV load was 709.3 IU/mL, ranging from 1.66 to 25,538.24 IU/mL. Among the 31 real time PCR positives for the presence of HBV DNA, 4 samples were HBsAg negative by the Alere Determine rapid diagnostic test kit. During the 4- to 6-month follow-up sampling, 48 patients were lost to follow up, 51 out of 65 patients whose blood samples were obtained tested negative on real-time PCR for HBV DNA. Out of these 51, 4 had their baseline samples testing positive (Table 5). Among the follow up samples, 14 had detectable DNA with 1 having a negative test results at baseline. Seven out of the 14 (50%) follow up samples with detectable DNA were quantitatively positive with a median VL of 223.62 IU/mL (IQR = 0.41–11890 IU/mL) (Table 5). Out of the seven samples with quantifiable HBV DNA, 3 had an increased HBV VL whilst four were decreased (Table 5).

## Prevalence of HDV in HIV/HBV co-infections among study population

To determine the exposure of HIV/HBV co-infected patients to HDV, the level of immuno-logic response to HDV was checked by measuring the levels of immunoglobulin G (IgG) in

**Table 4. Characteristics of HIV status of patients.**

| HIV Status | Number of Patients | Viral Load (cp/mL) |
|---|---|---|
| YES | 113 | |
| NO | 0 | |
| **HIV RNA Copies (cp/mL)** | | |
| **Baseline Samples** | 113 | |
| Target not detected | 47 | |
| <20 | 28 | |
| >20 | 36 | |
| Failed | 2 | |
| Median of >20, (IQR) | | 3054.5 (21–1265314) |
| **Follow Up** | | |
| Lost to follow up | 48 | |
| Target not detected | 26 | |
| Sustained VL | 20 | |
| Reduced VL | 6 | From <20 and 62 (BL) to Undetectable limit (FU) |
| <20 copies | 14 | |
| Sustained VL | 4 | |
| Reduced VL | 3 | From 23, 33 & 197 to <20 |
| Increased VL | 7 | From undetectable VL (BL) to <20 |
| >20 cp/mL | 22 | |
| Increased VL | 15 | To a median of 5635 (22–493032) |
| Reduced VL | 7 | From 55997 (41–706880) BL to 71 (29–7441) FU* |
| Median of >20 copies, (IQR) | | 111 (22–493032) |

* = Median (IQR); VL = Viral load; IQR = Interquartile range, AVE = Average.

the patient serum sample. Of the 113 serum samples tested, 2 (1.8%) had serological indication of exposure to HDV. The presence of HDV RNA was also tested in all the patient samples; both baseline and follow-up samples, with 4 testing positive (comprising 2 ELISA-positive and 2 ELISA negative samples). Thus, resulting in an overall prevalence of HDV in HIV/HBV co-infected patients of 3.5% in the study population (Fig 2).

## Biochemical analysis of ALT and AST in HIV-infected patients co-infected with HBV and HDV

To assess the biochemical presentation of HBV and HDV in HIV/HBV CO-infected individuals, firstly, a comparison of the liver enzymes (ALT and AST) levels between HBsAg negative and positive patients were done. There were no statistically differences in liver enzymes when a comparison between HBsAg positive and HBsAg negative patients as well as HDV positive patients and HDV negative patients was done.

To test whether the amount of HBV DNA affects the level of liver enzymes activity, a univariate logistic regression analysis of HBV DNA and the various risk factors such as Age, HIV load, infection type and ART regimen affecting the liver enzyme activity was tested. The patients who had low HBV DNA were observed to have low levels of ALT compared to those whose HBV DNA were high (p-value = 0.027) (Table 6). For AST none of the variables were significant except for age which odds ratio analysis showed to increase the risk of hepatic decompensation by 12% (OR = 1.125, CI = 1.0099–1.253, p-value = 0.032). (Table 7).

All four HDV positive patients have lamivudine (3TC) as part of their ART regimen with one patient on both lamivudine and tenofovir (TDF) based regimen. High ALT (51 IU/mL)

**Table 5. Characteristics of HBV status of patients.**

| HBV Status | Number of Patients | Viral Load (IU/mL) |
|---|---|---|
| **RDT** | | |
| Yes | 100 | - |
| No | 13 | - |
| **ECLIA** | | |
| Median, IU/mL (IQR) | 76 | 1934 (48–9118) |
| **Real time PCR (Baseline)** | 113 | |
| **Negative** | 82 | - |
| **Positive** | 31 | - |
| Qualitative | 18 | - |
| Quantitative | 13 | 709 (1.66–25538.25) |
| **Occult HBV Infection** | 4 | |
| **Real time PCR (Follow up)** | 64 | |
| **Negative** | 50 | - |
| **Positive** | 14 | - |
| Qualitative | 7 | - |
| Quantitative | 7 | 709 (1.66–25538.25) |
| Increased VL | 3 | (3704.32 BL– 13498.69 FU)* |
| Decreased VL | 4 | (3474.58 BL– 2447.979 FU)* |

* = Average of the viral load of the number of samples counted; BL = Baseline; FU = Follow up; VL = Viral load; IQR = Interquartile range.

and AST (43 IU/mL) levels were observed in the HDV positive patient receiving TDF as part of the ART regimen as compared to HDV positive patients not receiving TDF (Fig 3).

Comparative presentation of level of liver enzymes in HDV positive patients receiving TDF against HDV positive patients who are on only lamivudine-based ART regimen. No statistical analysis could be done as there is only one patient out of the 4 HDV+ patients who was on TDF.

## HBV and HDV genotyping

Of the 31 HBV positive samples identified by real time PCR, amplification of the HBV S-gene and nucleotide sequencing were successful for 15 of the samples. After aligning the 15 generated sequences with different genotypes of HBV reference strains from GenBank, all the samples clustered with reference strains of genotype E (Fig 4).

All 15 HBV sequences generated in this study have been deposited into the GenBank with the following accession numbers: HD-007 MN996900, HD-010 MN996901, HD-16 MN996902, HD-026 MN996903, HD-028 MN996904, HD-035 MN996905, HD-047 MN996906, HD-071 MN996907, HD-080 MN996908, HD-086 MN996909, HD-088 MN996910, HD-089 MN996911, HD-091 MN996912, HD-094 MN996913, HD-098 MN996914.

All samples clustered with genotype E reference strains from Ghana and Senegal. Sequences from this work are highlighted in red fonts.

Using primers targeting the R0 region of the HDV genome, all the 4 HDV-positive samples were amplified successfully, but nucleotide sequences could only be obtained for 2 samples. The sequenced samples were aligned with different genotypes of HDV reference strains from GenBank and all clustered with Genotype 1 reference strains as shown in Fig 5.

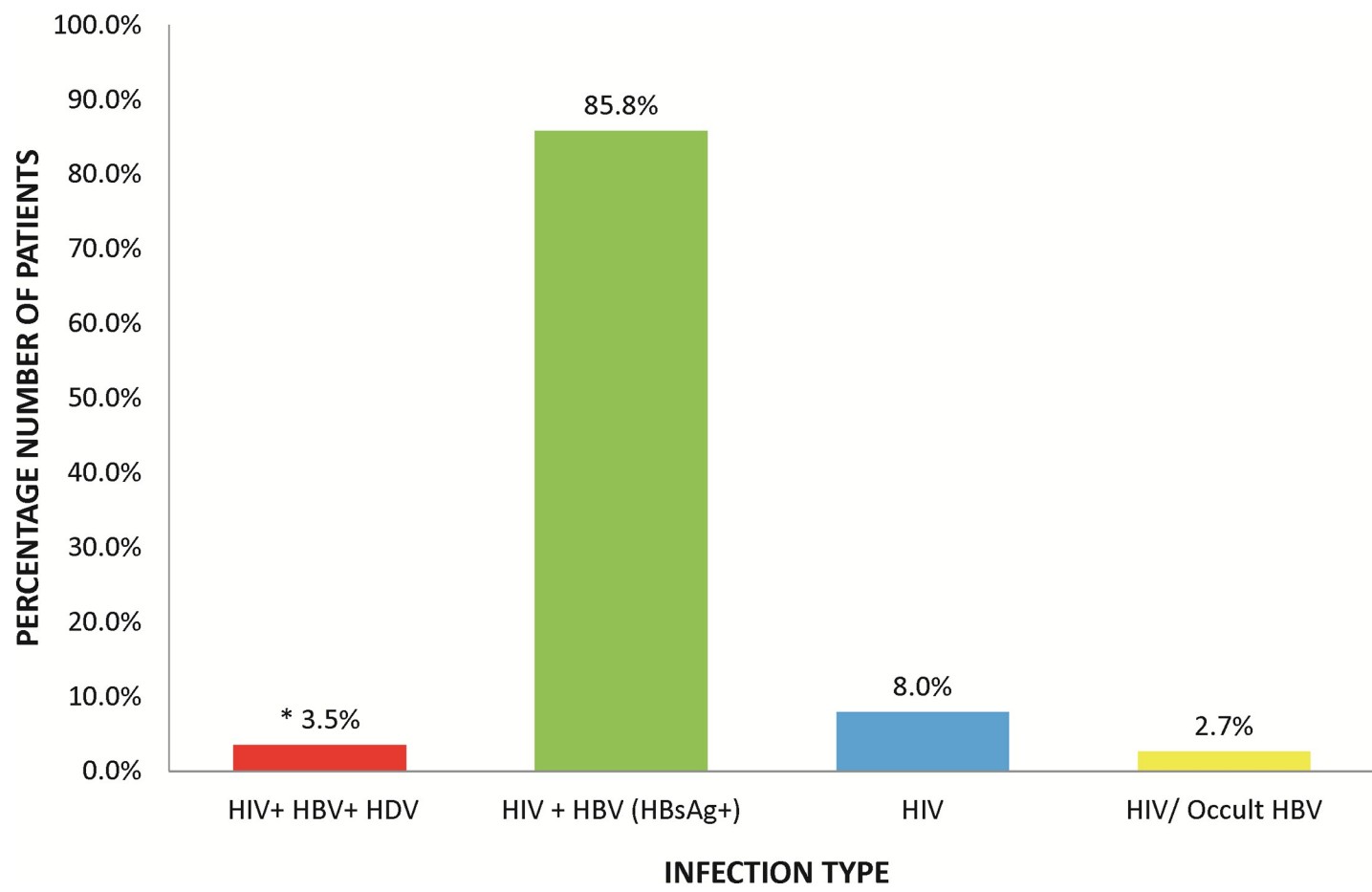

**Fig 2. Percentage distributions of infection types in enrolled patients.** *One patient had HIV/occult HBV infection. Occult HBV infection results from detection of HBV DNA in the serum of an HBsAg seroconverted patient.

**Table 6. Univariate analysis between liver decomposition between various risk factors.**

| Independent Variables | | Univariate Analysis | | |
|---|---|---|---|---|
| Variable | Category | High ALT | Low ALT | Test Statistic |
| | | [n = 13(11.50%)] | [n = 100(89.50)] | (p-value) |
| | | 48 (35–93) | 17 (2–34) | |
| ART regimen | With TDF | 7 (70.0) | 68 (71.58) | $X^2 = 0.01$ (0.916) |
| | Without TDF | 3 (30.0) | 27 (28.42) | |
| HBV VL Suppress | Yes | 5 (71.43) | 69 (94.52) | $X^2 = 4.91$ (0.027) |
| | No | 2 (28.57) | 4 (5.48) | |
| HBV/HIV co-infect | Reactive | 11 (84.62) | 89 (89) | $X^2 = 0.22$ (0.641) |
| | Nonreactive | 2 (15.38) | 11 (11) | |
| Sex | Male | 8 (61.54) | 42 (42) | $X^2 = 1.78$ (0.182) |
| | Female | 5 (38.46) | 58 (58) | |
| Age | Mean | 49 | 44 | t = 1.86(0.06) |

NB: Low ALT $<35$ IU/ml; High ALT $\geq 35$IU/ml.

**Table 7. Univariate analysis between liver decomposition between various risk factors.**

| Independent Variables | | Univariate Analysis | | |
|---|---|---|---|---|
| Variable | Category | High AST | Low AST | Test Statistic |
| | | [n = 13(12.38%)] | [n = 92(87.62%)] | (p-value) |
| ART regimen | With TDF | 8 (61.54) | 67 (72.83) | $X^2$ = 0.71 (0.399) |
| | Without TDF | 5 (38.46) | 25 (27.17) | |
| HBV VL Suppress | Yes | 9 (0.00) | 65 (91.55) | $X^2$ = 0.8222 (0.365) |
| | No | 0 (100.00) | 4 (8.45) | |
| HBV/HIV co-infect | Reactive | 12 (85.71) | 88 (88.89) | $X^2$ = 0.12 (0.728) |
| | Non reactive | 2 (14.29) | 11 (11.11) | |
| Sex | Male | 9 (64.29) | 41 (41.41) | $X^2$ = 1.78 (0.182) |
| | Female | 5 (35.71) | 58 (58.59) | |
| Age | Mean | 50 | 44 | t = 2.34(0.01)* |

NB: Low AST <35 IU/ml; High AST ≥35 IU/ml.

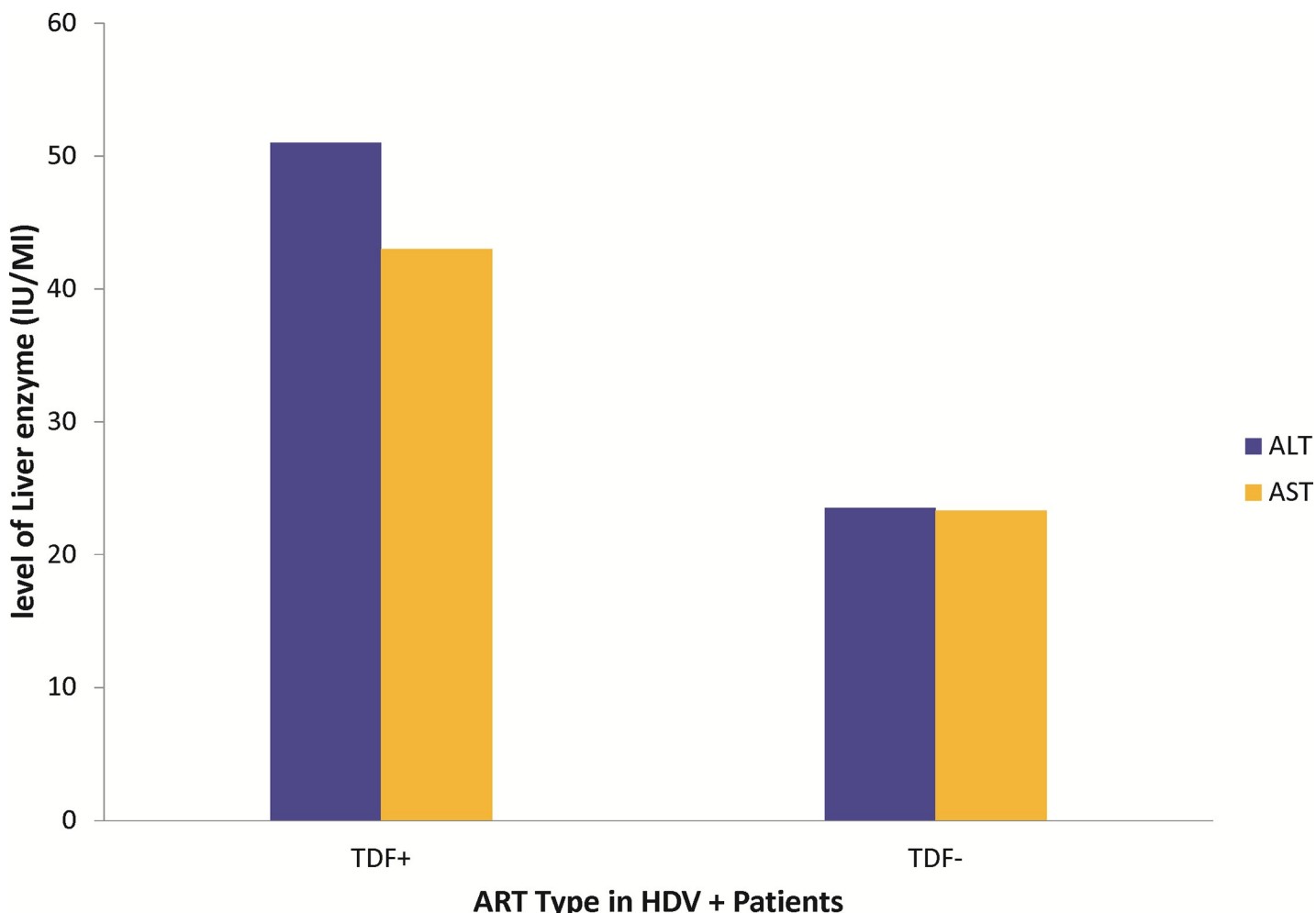

**Fig 3. Effect of TDF on ALT and AST.**

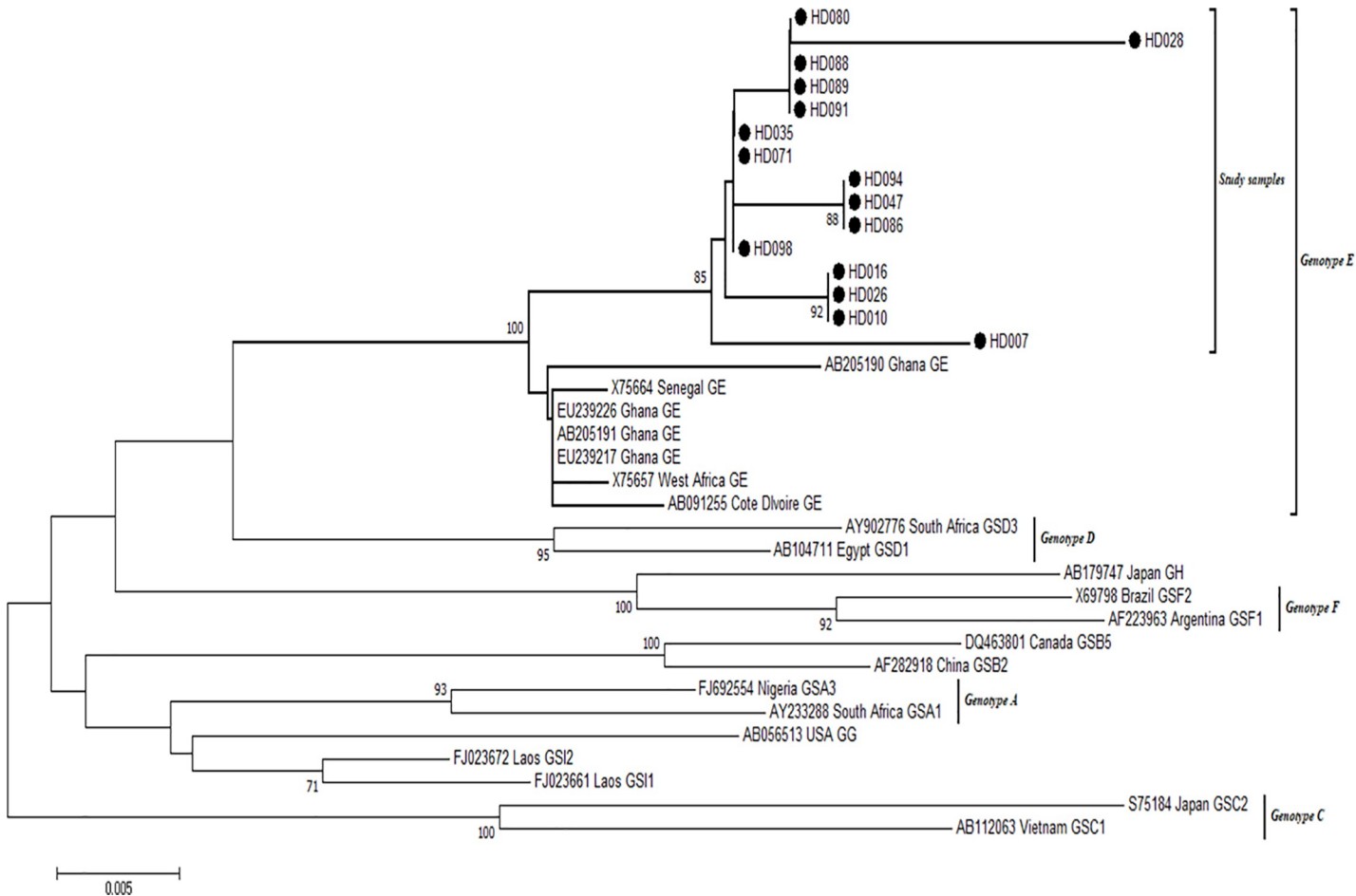

**Fig 4. Phylogenetic analysis of HBV sequences.** The alignment reconstructed phylogenies included 15 HBV study sample sequences (marked with black-shaded circles) as well as selected sequences available from the GenBank by April 2020.

## Discussion

Globally, an estimated 15 million HBV infected people are co-infected with HDV [16]. Although people infected with HDV have been shown to experience severe forms of viral hepatitis, HDV is often not tested for and thus considered as a neglected virus [17]. In Ghana, about 13.6% of HIV patients are HBV infected [18] with little information on the presence of HDV. Therefore, this study sought to determine the presence and genotype of HDV, as well as, the genotype of HBV among HIV/HBV co-infected patients in Ghana.

The 2% anti-HDV IgG antibodies detected in our study patients is consistent with the 2% seroprevalence results obtained for HIV/HBV co-infected patients in Ghana by Stockdale and colleagues [19]. Meanwhile, because anti-HDV antibody production might delay by several months after onset of infection, all patient samples were tested for HDV RNA despite their ELISA results, leading to an overall prevalence of about 4% (i.e. 4 samples) in this study. Also, all 4 HIV/HBV/HDV samples were reactive to the HIV confirmatory test and had detectable HDV RNA, but 3 out of the 4 samples had undetectable HBV DNA even though they were HBsAg reactive. The inability to detect HBV DNA could have been as a result of the HDV hijacking the HBV machinery to replicate itself with little or no effect on HBsAg production [20–23]. Besides, the effect of antiretroviral drugs to abate HBV replication cannot be

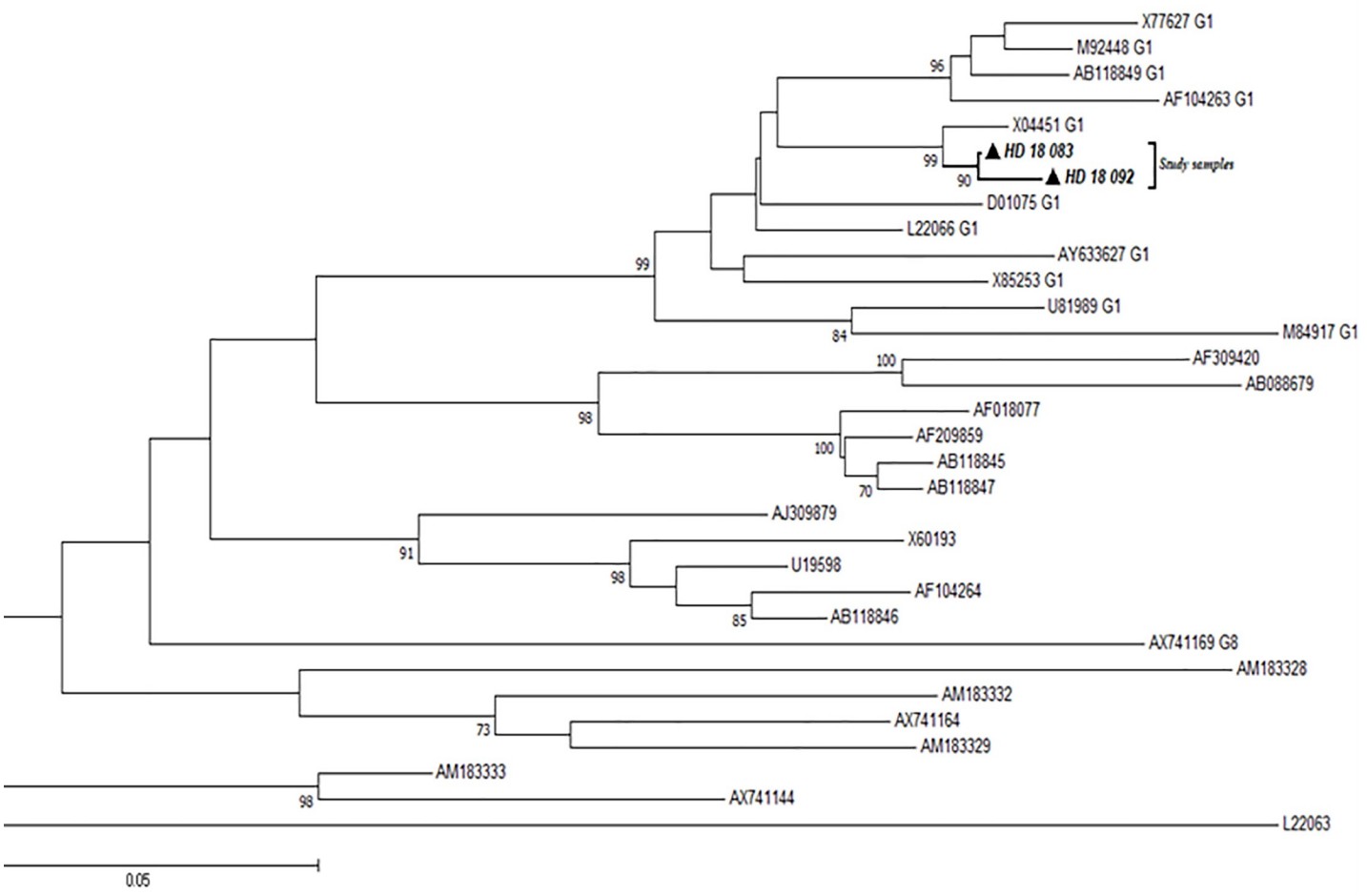

**Fig 5. Phylogenetic analysis of HDV sequences.** The alignment reconstructed phylogenies included 2 HDV study sample sequences (marked with black-shaded triangles) as well as selected sequences available from the GenBank by April 2020.

overlooked, as 70% of the HBsAg positive patients also had negative HBV DNA results. This result is justified by another study reporting a positive effect of TDF on HDV [24].

An increase or decrease in the prevalence of HIV/occult HBV infection has similarly been reported [25, 26]. More so, occult HBV infection could have an association with progression to liver cancer and cirrhosis in HIV infected patients [27–30]. Occult HBV infection also increases the risk of HBV infection in blood recipients during blood transfusion, since the main screening strategy deployed for HBV detection is HBsAg evaluation.

Amongst the HIV/HBV co-infected patients receiving highly active antiretroviral therapy (HAART), elevated liver enzymes can be related to toxicity of the anti-retroviral drugs (three- to five-fold) which may cause injury to the liver through direct or indirect effect [31–37]. The effect of ART on the liver enzyme in our study showed no association to AST (p-value = 0.916). This observation is consistent with a study by Soriano et al. in 2014 [24]. Hepatic flares in ALT and AST levels have been observed to be related to HBeAg seroconversion in chronic Hepatitis B patients. A study observed high HBV DNA and lower liver enzyme elevations with more cirrhosis [38] whilst another study correlated HBV DNA levels with serum ALT to be significant suggesting that patients whose HBV DNA levels are elevated have an increase in liver inflammation [39]. In our study, we noted that patients with baseline HBV DNA greater than 2000 IU/mL had normal to low levels of ALT and AST. The levels of ALT and AST

reduced during follow-up testing and a univariate logistic regression analysis of HBV DNA with ALT and AST levels revealed that suppression of HBV can be correlated with low levels of ALT, consistent with earlier submission [38]. HBV positive patients are at risk of getting infected with hepatitis D virus which also affects ALT/AST levels [38]. However, in our study we noticed that infection type does not necessarily affect the levels of ALT or AST. Thus, whether a patient had HDV/HIV/HBV or HIV/HBV, or HIV alone did not have significant association to the ALT and AST (p-value > 0.05).

Furthermore, genetic characterisation of patients HBV yielded genotype E, all of which clustered with West African sequences–including Ghana, as confirmed by earlier reports [40–42], where genotype E was identified as the predominant with low numbers of genotypes A and D.

We also observed in our work that, two out of four positive HDV samples were determined to be genotype 1. This corroborates findings of HDV genotype 1 among HBV mono-infected patients in Korle-Bu, Ghana (unpublished data).

As a result of the few number of HDV RNA positives obtained as well as the loss to follow up samples amongst which the HDV positive samples were part made it not possible to study the effect HDV has on HBV. The reason given above also made it not possible to look at the effect of HDV on liver disease progression as some studies [3, 4] have shown HDV to increase liver disease progression in patients co-infected or super-infected with HDV.

## Conclusions

Data obtained in our research work established the occurrence of HDV in patients co-infected with HIV/HBV to be 3.54% among our study population. No correlation was observed between ALT/AST levels and other risk factors such as age, gender, type of ART, duration of ART, and HIV load. The incidence of occult hepatitis B infection, which is an essential factor in the development of cirrhosis and liver cancer, was also identified. We observed that HBV DNA suppression, correlated with low levels of ALT whilst no association was observed between HBV DNA suppression and AST.

With the presence and circulation of HDV and incidence of occult HBV infection in HIV/HBV co-infected patients in Ghana, it is vital to look out for the presence of HDV and occult HBV in HIV/HBV co-infected patients especially during screening of blood being donated for transfusion purposes as these may present potential risk of liver cancers, HBV and HDV transmission.

Amongst the HIV/HBV co-infected patients recruited for this study, HBV genotype E and HDV genotype 1 were observed to be in circulation.

Based on the findings of our research, we recommend the conduction of further studies that would assess the clinical outcomes of both HDV genotype 1 and HBV genotype E cases among HIV/HBV in Ghana.

## Supporting information

**S1 Questionnaire.**
(PDF)

## Acknowledgments

We are grateful to all the patients who participated in this study, the staff of the Fever's Unit, and staff of the Virology Department at NMIMR and Biochemistry, Cell and Molecular Biology.

## Author Contributions

**Conceptualization:** Keren Attiku, Joseph Bonney, Peter Puplampu, Osbourne Quaye.

**Data curation:** Keren Attiku, Joseph Bonney, Esinam Agbosu, Evelyn Bonney, John Odoom, James Aboagye, Seth Agyemang, Yaw Awuku-Larbi, Augustina Arjarquah, Gifty Mawuli, Osbourne Quaye.

**Formal analysis:** Keren Attiku, Joseph Bonney, Evelyn Bonney, John Odoom, James Aboagye, John Mensah, Yaw Awuku-Larbi, Augustina Arjarquah, Gifty Mawuli, Osbourne Quaye.

**Funding acquisition:** Keren Attiku.

**Investigation:** Keren Attiku, Joseph Bonney, Esinam Agbosu, Peter Puplampu, Vincent Ganu, John Mensah, Seth Agyemang, Gifty Mawuli, Osbourne Quaye.

**Methodology:** Keren Attiku, Joseph Bonney, Evelyn Bonney, Peter Puplampu, Vincent Ganu, James Aboagye, Seth Agyemang, Augustina Arjarquah.

**Project administration:** Keren Attiku, Joseph Bonney.

**Supervision:** Joseph Bonney, Evelyn Bonney, John Odoom, Osbourne Quaye.

**Visualization:** Keren Attiku, Joseph Bonney, James Aboagye.

**Writing – original draft:** Keren Attiku, Joseph Bonney, Esinam Agbosu, Evelyn Bonney, Peter Puplampu, Vincent Ganu, John Odoom, James Aboagye, John Mensah, Seth Agyemang, Yaw Awuku-Larbi, Augustina Arjarquah, Gifty Mawuli, Osbourne Quaye.

**Writing – review & editing:** Keren Attiku, Joseph Bonney, Esinam Agbosu, Evelyn Bonney, Peter Puplampu, Vincent Ganu, John Odoom, James Aboagye, John Mensah, Seth Agyemang, Yaw Awuku-Larbi, Augustina Arjarquah, Gifty Mawuli, Osbourne Quaye.

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
