## [Decision Letter · Decision Letter 0]

3 Sep 2020

PONE-D-20-18263

Circulation of hepatitis delta virus and occult hepatitis B virus infection amongst HIV/HBV co-infected patients in Korle-Bu, Ghana.

PLOS ONE

Dear Dr. Bonney,

Thank you for submitting your manuscript to PLOS ONE. After careful consideration, we feel that it has merit but does not fully meet PLOS ONE’s publication criteria as it currently stands. Therefore, we invite you to submit a revised version of the manuscript that addresses the points raised during the review process.

We look forward to receiving your revised manuscript.

Kind regards,

Jason Blackard, PhD

Academic Editor

PLOS ONE

Journal Requirements:

2. Please provide additional details regarding participant consent. In the ethics statement in the Methods and online submission information, please ensure that you have specified (1) whether consent was informed and (2) what type you obtained (for instance, written or verbal, and if verbal, how it was documented and witnessed).

3. Please address the following:

- Please include additional information regarding the survey or questionnaire used in the study and ensure that you have provided sufficient details that others could replicate the analyses. For instance, if you developed a questionnaire as part of this study and it is not under a copyright more restrictive than CC-BY, please include a copy, in both the original language and English, as Supporting Information.

- Please ensure you have thoroughly discussed any potential limitations of this study within the Discussion section, including the limited sample size.

Additional Editor Comments (if provided):

This is a longitudinal study of hepatitis D infection in individuals with HIV/HBV co-infection in Ghana.  Given the high burden of HBV and the sparse data on HDV co-infection in sub-Saharan Africa, studies like this are important.

The overall study number (n = 113) is moderate in size; however, the exclusion of previously vaccinated individuals is a good idea, as is the longitudinal study design.

There are a few awkward phrases and formatting mistakes.  The manuscript would benefit from careful review by a native English speaker and/or a professional editing service.

The lower limits of detection for the HBV DNA and HDV RNA assays should be stated explicitly.

It is also important to report the specific regions of HBV and HDV being amplified by these assays.

Additional details about the S gene PCR should be provided, including the nucleotide ranges of the primers and the size of the S PCR fragment analyzed.

Similarly, the same details should be added for the HDV RNA PCR.

While most HIV/HBV co-infected individuals received ART, how many were receiving ART regimens containing lamivudine or tenofovir?  This is not explicitly stated in the methods or the results.

The HDV prevalence results presented in lines 206-212  are a bit confusing.  Two samples were ELISA positive but were an additional two samples HDV RNA positive but not ELISA positive?

What is the definition of low versus high ALT or AST used in Table 4?

The low PCR positivity rate (15 of 31) suggests that the PCR is not optimized.  It is also important to try other primer sets that amplify a larger portion of the HBV genome or even attempt full-length PCR.  When using multiple primer sets, the PCR positivity rate increases substantially.

Is it correct that all 4 HDV positive samples could be amplified by RT-PCR but only two quality sequences were obtained?

Study limitations should be discussed including the small population size and the inability to genotype all samples.

It is unclear if HDV seroprevalence or characterization of HDV genotypes have ever been reported in Ghana?  What about neighboring countries in West Africa?

For the two phylogenetic trees, showing on relevant bootstrap values (typically those >70%) would be best.  Of note, there are more robust ways to generate a phylogenetic tree . . . consider Bayesian inference.

For the HBV genotype E sequences, the interpatient genetic distances should be reported so that the reader has some understanding of how similar or different these sequences are.

Reviewers' comments:

Reviewer's Responses to Questions

**Comments to the Author**

1. Is the manuscript technically sound, and do the data support the conclusions?

Reviewer #1: Yes

Reviewer #2: Partly

2. Has the statistical analysis been performed appropriately and rigorously? 

Reviewer #1: Yes

Reviewer #2: Yes

3. Have the authors made all data underlying the findings in their manuscript fully available?

Reviewer #1: Yes

Reviewer #2: Yes

4. Is the manuscript presented in an intelligible fashion and written in standard English?

Reviewer #1: Yes

Reviewer #2: Yes

5. Review Comments to the Author

Reviewer #1: Attiku et al reported in the manuscript the prevalence of HDV and occult HBV infection in a cohort of 113 HBV/HIV infected patients in Korle-Bu, Ghana. The study is interesting and addressed important association of HDV and occult HBV in HBV/HIV coinfected patients in Ghana. The authors concluded that HDV prevalence in this cohort is about 3.45% and the occult HBV is about 30.8%. The study is well-designed and the manuscript is well-written. The following suggestions and information are needed to support the conclusion of the study:

1. What is the sensitivity and the specificity of the tests involved in the study such as the HBV DNA, the HBsAg tests, HDV antibody tests and HBV DNA tests.

2. In the longitudinal follow up study, what is the incidence of OCCULT HBV, and HDV?

Reviewer #2: Overall, the manuscript is well written the main points are clearly stated. The data presented is important in showing the extent of HDV and also the extent of occult HBV among HIV/HBV co-infected patients. However, a few comments for improvements are suggested

1. If all the 113 study samples were HIV/HBV co-infected yet only 100 were HBV positive, on what basis are the other 13 samples considered HBV/HIV co-infected? This is moreso since in the methods it was indicated that those who were HIV/HBV negative were excluded (line 87) and additionally in the results section it is indicated “All the patients recruited were confirmed by serological analyses to be HIV/HBV co-infected” (line 147/148).

2. The methods used to assay enzyme levels have not been included

3. What was the rationale of follow up samples in this study considering the high rate of loss to follow up? I am persuaded that the findings of this study would not largely change even if these follow up samples were not collected.

4. Could the authors provide a definition of what does low ALT/high ALT, high HBV DNA mean in line 226/227?

5. Table 4 only presents ALT as a marker of liver decomposition. What was the rationale of ommiting the AST results??

6. PLOS authors have the option to publish the peer review history of their article (what does this mean?). If published, this will include your full peer review and any attached files.

Reviewer #1: **Yes: **Mohamed Tarek Shata

Reviewer #2: No

---

## [Author Response · Author response to Decision Letter 0]

15 Oct 2020

PONE-D-20-18263 

"Circulation of hepatitis delta virus and occult hepatitis B virus infection amongst HIV/HBV co-infected patients in Korle-Bu, Ghana" 

Letter responding to comments:

We are pleased that PLOS ONE will consider a review of our manuscript pending satisfactory revisions as suggested by the editor-in-chief and the reviewers. We are grateful to you for your constructive comments. We explain in detail below how we have addressed the concerns and have indicated the page and lines of the revised text in the accompanying revised manuscript.

Review Comments

Comment Author Response Lines

 The authors have duly complied to ensure that all requirements are met. 

2. Please provide additional details regarding participant consent. In the ethics statement in the Methods and online submission information, please ensure that you have specified (1) whether consent was informed and (2) what type you obtained (for instance, written or verbal, and if verbal, how it was documented and witnessed).

 Written informed consent was obtained from eligible patients. This has been addressed. 90

3. Please address the following:

- Please include additional information regarding the survey or questionnaire used in the study and ensure that you have provided sufficient details that others could replicate the analyses. For instance, if you developed a questionnaire as part of this study and it is not under a copyright more restrictive than CC-BY, please include a copy, in both the original language and English, as Supporting Information.

- Please ensure you have thoroughly discussed any potential limitations of this study within the Discussion section, including the limited sample size.

 The questionnaire that was used for data collection has been added as a supporting information.

A limitation of the study has been added as the last paragraph of the Discussion 

334 - 338

 The abstract in the online submission form and the revised manuscript have been amended accordingly. 

 Lines 29 to 57

5. There are a few awkward phrases and formatting mistakes. The manuscript would benefit from careful review by a native English speaker and/or a professional editing service.

 The authors have carefully read through the revised manuscript and corrected wrong phrases and formatting mistakes. 

6. The lower limits of detection for the HBV DNA and HDV RNA assays should be stated explicitly.

 The lower limits of detection for both assays have been stated.

 126 - 127

7. It is also important to report the specific regions of HBV and HDV being amplified by these assays.

 The specific regions that were amplified in both HBV and HDV are the core genes. This information has been added to the Methods section. 

 114 - 115

8. Additional details about the S gene PCR should be provided, including the nucleotide ranges of the primers and the size of the S PCR fragment analyzed.

 Details about the S gene, and nucleotide ranges of the primers and the size of the PCR fragment analyzed have been added to the Methods section. 132 - 134

9. Similarly, the same details should be added for the HDV RNA PCR.

 Details about the 3’ RO gene, nucleotide ranges of the primers, and the size of the PCR fragment analyzed have been added to the Methods section. 146 - 150

10. While most HIV/HBV co-infected individuals received ART, how many were receiving ART regimens containing lamivudine or tenofovir? This is not explicitly stated in the methods or the results.

 The number of individuals who are on lamivudine-based or tenofovir-based or both ART have been stated in the Results section.

 175 - 176

11. The HDV prevalence results presented in lines 206-212 are a bit confusing. Two samples were ELISA positive but were an additional two samples HDV RNA positive but not ELISA positive?

 Yes, two samples were ELISA positive but additional two samples were HDV RNA positive that were not initially ELISA positive.

 231 - 232

12. What is the definition of low versus high ALT or AST used in Table 4?

 A legend with cut-off values has been added to Tables 6 and 7 to explain low and high ALT and AST.

 254, 256

13. The low PCR positivity rate (15 of 31) suggests that the PCR is not optimized. It is also important to try other primer sets that amplify a larger portion of the HBV genome or even attempt full-length PCR. When using multiple primer sets, the PCR positivity rate increases substantially.

 The suggestion has been noted and will be used in subsequent studies.

14. Is it correct that all 4 HDV positive samples could be amplified by RT-PCR but only two quality sequences were obtained?

Even though four HDV positive samples were amplified by RT-PCR, only two quality sequences were obtained. Yes, that was what we experienced in this study.

15. Study limitations should be discussed including the small population size and the inability to genotype all samples.

 As per an earlier comment, a limitation of the study has been added as the last paragraph of the Discussion. 334 - 338

16. It is unclear if HDV seroprevalence or characterization of HDV genotypes have ever been reported in Ghana? What about neighboring countries in West Africa?

 HDV seroprevalence was mentioned in the Discussion section of this study. However, this is the first report on genotypic characterization of HDV in Ghana, apart from the unpublished data that was mentioned in the Discussion section. 293 - 295

17. For the two phylogenetic trees, showing on relevant bootstrap values (typically those >70%) would be best. Of note, there are more robust ways to generate a phylogenetic tree . . . consider Bayesian inference. The bootstrap values that are less than 70% have been removed from the phylogenetic trees.

 Figure 4 and Figure 5

18. For the HBV genotype E sequences, the interpatient genetic distances should be reported so that the reader has some understanding of how similar or different these sequences are. The authors do appreciate and consider this suggestion useful in measuring the genetic divergence between the genotype E sequences. We did not determine the interpatient genetic distances as was not included in the scope of work carried out in this study. We hope to include it in our subsequent related studies. 

REVIEWER 1

Comment Author Response Lines

1. What is the sensitivity and the specificity of the tests involved in the study such as the HBV DNA, the HBsAg tests, HDV antibody tests and HBV DNA tests The sensitivities and specificities as provided by the manufacturing companies for the various test kits have been provided in the Methods section.

 111 – 113, 128- 129

2. In the longitudinal follow up study, what is the incidence of OCCULT HBV, and HDV? There was no occult HBV nor HDV infections in the follow-up samples.

REVIEWER 2

Comment Author Response Lines

1. If all the 113 study samples were HIV/HBV co-infected yet only 100 were HBV positive, on what basis are the other 13 samples considered HBV/HIV co-infected? This is more so since in the methods it was indicated that those who were HIV/HBV negative were excluded (line 87) and additionally in the results section it is indicated “All the patients recruited were confirmed by serological analyses to be HIV/HBV co-infected” (line 147/148).

 All the 113 participants were coinfected based on their medical records. However, the participants were retested after recruitment for confirmation of their coinfection status. It was at the confirmatory testing that 13 of the patients were found to be only HIV positive. The sentence on lines 147 and 148 has be changed to “All the patients recruited were retested by serological analyses for their coinfection status”.

 166 - 167

2. The methods used to assay enzyme levels have not been included

 The assays that were used to determine the enzyme levels have been included in the Methods section.

 104 - 107

3. What was the rationale of follow up samples in this study considering the high rate of loss to follow up? I am persuaded that the findings of this study would not largely change even if these follow up samples were not collected. The rational for the follow-up was to determine the effect of HDV on HBV, which will also give an indication of the severity of liver diseases progression. 

4. Could the authors provide a definition of what does low ALT/high ALT, high HBV DNA mean in line 226/227?

 The cut-off values for ALT and HBV DNA have been added to the Results section. 254, 256

5. Table 4 only presents ALT as a marker of liver decomposition. What was the rationale of omitting the AST results?? The AST results was omitted because there was no association with the risk factors except age for the viral infections. A table (Table 7) for AST results have been included in the results section. 255

---

## [Decision Letter · Decision Letter 1]

9 Nov 2020

PONE-D-20-18263R1

Circulation of hepatitis delta virus and occult hepatitis B virus infection amongst HIV/HBV co-infected patients in Korle-Bu, Ghana.

PLOS ONE

Dear Dr. Bonney,

Thank you for submitting your manuscript to PLOS ONE. After careful consideration, we feel that it has merit but does not fully meet PLOS ONE’s publication criteria as it currently stands. Therefore, we invite you to submit a revised version of the manuscript that addresses the points raised during the review process.

Please address the minor issues raised by reviewer #2 prior to acceptance of the revised manuscript.

We look forward to receiving your revised manuscript.

Kind regards,

Jason Blackard, PhD

Academic Editor

PLOS ONE

Additional Editor Comments (if provided):

Please address the minor issues raised by reviewer #2 prior to acceptance of the revised manuscript.

Reviewers' comments:

Reviewer's Responses to Questions

**Comments to the Author**

1. If the authors have adequately addressed your comments raised in a previous round of review and you feel that this manuscript is now acceptable for publication, you may indicate that here to bypass the “Comments to the Author” section, enter your conflict of interest statement in the “Confidential to Editor” section, and submit your "Accept" recommendation.

Reviewer #1: All comments have been addressed

Reviewer #2: All comments have been addressed

2. Is the manuscript technically sound, and do the data support the conclusions?

Reviewer #1: Yes

Reviewer #2: Yes

3. Has the statistical analysis been performed appropriately and rigorously? 

Reviewer #1: Yes

Reviewer #2: Yes

4. Have the authors made all data underlying the findings in their manuscript fully available?

Reviewer #1: Yes

Reviewer #2: Yes

5. Is the manuscript presented in an intelligible fashion and written in standard English?

Reviewer #1: Yes

Reviewer #2: Yes

6. Review Comments to the Author

Reviewer #1: The authors responded adequately to the comments of the reviewers, and addressed my concern satisfactory.

Reviewer #2: I find some inconsistencies in the results presented in Table 3 on Demographic Characteristics of Study Participants regarding ART. Page 9 line 176 states that two patients were ART naive (out of the 113 study participants). However, in Table 3, under duration of ART, a total of 112 patients are indicated. Based on the information, it would have been expected the total number of participants on ART would be 111. This is further confirmed by the data on class of ART in Table 3 which indicates 111 study participants. Could the authors explain this anomaly.

7. PLOS authors have the option to publish the peer review history of their article (what does this mean?). If published, this will include your full peer review and any attached files.

Reviewer #1: **Yes: **Mohamed Tarek Shata

Reviewer #2: **Yes: **Dr George Gachara

---

## [Author Response · Author response to Decision Letter 1]

9 Dec 2020

The authors are grateful to the reviewer for bringing this up. The correction has been done on Table 3 - Demographic Characteristics of Study Participants regarding ART and the figure 112 has been changed to 113 in the revised manuscript. It should be 113 and not 111 because the duration of 0 to 24 months for the ART will take care of the 2 patients who were naïve (zero duration).

---

## [Editor Report · Decision Letter 2]

11 Dec 2020

Circulation of hepatitis delta virus and occult hepatitis B virus infection amongst HIV/HBV co-infected patients in Korle-Bu, Ghana.

PONE-D-20-18263R2

Dear Dr. Bonney,

We’re pleased to inform you that your manuscript has been judged scientifically suitable for publication and will be formally accepted for publication once it meets all outstanding technical requirements.

Kind regards,

Jason Blackard, PhD

Academic Editor

PLOS ONE

Additional Editor Comments (optional):

None

Reviewers' comments:

None

---

## [Editor Report · Acceptance letter]

28 Dec 2020

PONE-D-20-18263R2 

Circulation of hepatitis delta virus and occult hepatitis B virus infection amongst HIV/HBV co-infected patients in Korle-Bu, Ghana. 

Dear Dr. Bonney:

I'm pleased to inform you that your manuscript has been deemed suitable for publication in PLOS ONE. Congratulations! Your manuscript is now with our production department. 

Kind regards, 

on behalf of

Dr. Jason Blackard 

Academic Editor

PLOS ONE